# A Generalized Binary Tree Mechanism for Private Approximation of All-Pair Shortest Distances

**Zongrui Zou, Jingcheng Liu**[*]
State Key Laboratory for Novel Software Technology, New Cornerstone Science Laboratory
Nanjing University
zou.zongrui@smail.nju.edu.cn, liu@nju.edu.cn

**Chenglin Fan**
Department of Computer Science and Engineering
Seoul National University
fanchenglin@gmail.com

**Michael Dinitz**[†]
Department of Computer Science
Johns Hopkins University
mdinitz@cs.jhu.edu

**Jalaj Upadhyay**[‡]
Management Science & Information Systems Department
Rutgers University
jalaj.upadhyay@rutgers.edu

## Abstract

We study the problem of approximating all-pair distances in a weighted undirected graph with differential privacy, introduced by Sealfon [Sea16]. Given a publicly known undirected graph, we treat the weights of edges as sensitive information, and two graphs are neighbors if their edge weights differ in one edge by at most one. We obtain efficient algorithms with significantly improved bounds on a broad class of graphs which we refer to as *recursively separable*. In particular, for any $n$-vertex $K_h$-minor-free graph, our algorithm achieve an additive error of $\widetilde{O}(h(nW)^{1/3})$, where $W$ represents the maximum edge weight; For grid graphs, the same algorithmic scheme achieve additive error of $\widetilde{O}(n^{1/4}\sqrt{W})$. Our approach can be seen as a generalization of the celebrated binary tree mechanism for range queries, as releasing range queries is equivalent to computing all-pair distances on a path graph. In essence, our approach is based on generalizing the binary tree mechanism to graphs that are *recursively separable*.

---

[*]JL and ZZ have been supported by National Science Foundation of China under Grant No. 62472212 and the New Cornerstone Science Foundation.

[†]Supported in part by NSF award 2228995

[‡]JU's research was funded by the NSF CNS 2433628, Google Seed Fund grant, Google Research Scholar Award, Dean Research Seed Fund, and Rutgers Decanal Grant no. 302918.

# 1 Introduction

Graph-structured data arises in an enormous number of settings. As a result, analysis of graphs has important applications in not just computer science (networking, search engine optimization, bioinformatics, machine learning, etc.), but also in biology, chemistry, transportation and logistics, supply-chain management, operations research, and even in everyday applications like finding the shortest route on a map using GPS navigation. Since in some applications these graphs can encode sensitive personal information, the analysis of sensitive *graph* data while preserving privacy has attracted growing attention in recent years. One of the main privacy notions that has been proposed and used extensively in such graph analysis is *differential privacy*, with successful applications including cut approximation [30, 24, 15, 39, 11], spectral approximation [6, 4, 44, 1], correlation or hierarchical clustering ([9, 14, 32, 13]) and numerical statistics release ([34, 43, 8, 17, 33]), among others.

Recently, starting with Sealfon [42], the problem of releasing *All-Pair Shortest Distances* (APSD) on a weighted graph with differential privacy has received significant attention [26, 25, 12, 23, 16, 7], due to the practical importance and applicability of private graph distances and the fact that the problem itself is inherently natural. In this line of work, given an $n$-vertex weighted graph $G = ([n], E, w)$, the goal is to output the values of the shortest distances between each pair of vertices, while preserving *weight*-level differential privacy, where two graphs are considered neighboring if they share the same edge set (i.e., the topology is public information) and differ in the weight of exactly one edge by at most one.

Sealfon [42] introduced the problem and use a post-processing scheme to gave an algorithm with $\widetilde{O}(n/\varepsilon)$ additive error for pure and approximate differential privacy with privacy budget $\varepsilon$. This bound was improved subsequently [12, 26] using a simple sub-sampling technique, reducing it to only $\widetilde{O}(\sqrt{n}/\varepsilon)$ for approximate differential privacy. For the lower bound side, the first result on private APSD was provided by Chen et al. [12], who established a lower bound of $\Omega(n^{1/6}/\varepsilon)$. This bound was recently improved to $\Omega(n^{1/4}/\varepsilon)$ by Bodwin et al. [7].

Though $\widetilde{O}(\sqrt{n})$ error is the best known for answering private APSD in general graphs, it is possible to significantly improve it for many special classes of graphs. For example, outputting all pairs shortest distances in an $n$-vertex *path* graph is equivalent to answering all possible range queries on a vector of length $n - 1$ [42]. Under weight-level privacy, this task can be solved with only polylog$(n)$ additive error using the classic *binary tree* mechanism [19, 10]. This mechanism leverages the Bentley and Saxe data structure [5] to reduce the number of compositions required for privacy. More generally, Sealfon [42] shows that the polylog$(n)$ error can also be achieved when privately answering the shortest distances in an $n$-vertex tree.

However, there is a lack of work on private APSD for graphs that lie between trees and general graphs, a gap that encompasses many graph classes commonly encountered in practical applications, such as *planar graphs*. Perhaps the most natural example of such an application is finding the shortest distances in road maps or metro networks, where all the routes can usually be represented in the plane, and edge weights are determined by traffic (higher edge weights indicate more traffic, which navigation tools should tend to avoid). Therefore, we have the following natural question:

> **Question 1.** *Is it possible to solve private all-pairs shortest distances with better utility in planar graphs (or other natural class of graphs) than in general graphs[4]?*

In this paper we answer this question in the affirmative, by showing that for any $K_h$-minor-free graph there exists an efficient and differentially private algorithm that achieves an additive error of $\widetilde{O}(h \cdot (nW)^{1/3})$ for answering APSD, where $W$ is the maximum edge weights. This result immediately implies an $\widetilde{O}((nW)^{1/3})$ error for private APSD on planar graphs, as every planar graph is $K_5$-minor-free [37]. Additionally, for a $\sqrt{n} \times \sqrt{n}$ grid graph, within the same framework, we further reduce the error of private APSD to only $\widetilde{O}(\frac{n^{1/4}W^{1/2}}{\varepsilon^{1/2}})$. These results improve multiple previous upper bounds that also depend on $W$. (See Section 1.1 for details.) Such an upper bound $W$

---

[4]On graphs with bounded tree-width $p$, an interesting result by [23] presents a purely differentially private algorithm that outputs all-pair shortest distances with an additive error of $\widetilde{O}(p^2/\varepsilon)$. This bound however, does not lead to an improvement on planar graphs, as the tree-width of a planar graph can be as large as $O(\sqrt{n})$.

on edge weights is also natural in real-world scenarios, such as packet routing, where it is defined by the bandwidth of the network.

From a technical point of view, our results can be considered as a generalization of the mechanism proposed in Sealfon [42] for computing APSD on an $n$-vertex tree. Their mechanism inherently extends the binary tree mechanism, which originally operates on a single path, to a tree structure, achieving an error bound of polylog($n$). Therefore, to further extend this approach to more general graph topologies beyond paths or trees, we pose the following fundamental question:

> **Question 2** *What underlying property allows the binary tree mechanism to reduce the error in answering APSD queries on paths or trees?*

To answer this question, we propose the concept of *recursive separability* (Definition 1). We show that the recursive separability of tree graphs is the inherent reason why the binary tree mechanism significantly reduces the error in answering APSD. Intuitively, a graph as *recursively separable* if it has a *balanced separator* such that when the separator is removed, the resulting subgraphs themselves are also recursively separable.

By working with recursively separable graphs, we develop a private algorithm based on a divide-and-conquer strategy to compute all-pair shortest distances, with an additive error characterized by the quality of the separators of the input graph. As a consequence, our algorithm not only encompasses the older use cases of the binary tree mechanism [19, 10, 20, 42], but also extends to a much broader class of graphs that exhibit recursive separability, including planar graphs or $K_h$-minor-free graphs. Importantly, our results also open up the opportunity to leverage a rich body of prior work on graph separation theorems (e.g. [38, 28, 3, 29, 2, 35, 27, 36]) to the design of private APSD algorithms.

## 1.1 Our Results

In this paper, we consider the standard *weight-level privacy* setting [42]. In this setting, two positive weighted $n$ vertices graphs $G = ([n], E, w)$, $G' = ([n], E, w')$ with the same edge topology are *neighboring* if $\|w - w'\|_0 \leq 1$ and $\|w - w'\|_1 \leq 1$, i.e., there is an edge which differ by at most 1. Here, $w, w' \in \mathbb{R}_{\geq 0}^{|E|}$ encodes the edge weights, where $\mathbb{R}_{\geq 0}$ denote the set of positive real numbers.

For a graph $G$, a *separator* of $G$ is a subset of vertices that, when removed, splits the graph into two or more disconnected components. Formally, a separator is a subset of vertices $S \subseteq V$ where there exist disjoint vertex sets $A$ and $B$ such that $A \cup B = V \backslash S$ and no edge joins a vertex in $A$ with a vertex in $B$. To privately compute all-pairs shortest distances using a divide-and-conquer approach, we first introduce the concept of recursive separability.

**Definition 1.** Fix $p \in \mathbb{N}$ and $q, q' \in \mathbb{R}$ such that $\frac{1}{2} \leq q < q' < 1$. An undirected graph $G = (V, E)$ is $(p, q, q')$-*recursively separable* if and only if either $|V| = O(1)$ or:

- $G$ has a separator $S \subseteq V$ of size $|S| \leq \min\{p, (q' - q)|V|\}$ such that each connected component of $G' = (V \backslash S, E(V \backslash S))$ has at most $q|V|$ vertices, and

- Every subgraph of $G$ is $(p, q, q')$-recursively separable.

**Remark 2.** We note that the definition of recursive separability is *oblivious* to edge weights. Thus, we say a weighted graph is recursively separable if its underlying graph is recursively separable.

Our first result is a general algorithm that privately computes all-pairs shortest distances for any graph, with an error bound that depends on the quality of the graph's separators.

**Theorem 3.** *Fix privacy budgets $0 < \varepsilon, \delta < 1$, and suppose $\frac{1}{2} \leq q < q' < 1$ are constants. For any weighted $n$-vertex $(p, q, q')$-recursively separable graph $G$, there exists an $(\varepsilon, \delta)$-differentially private algorithm such that with high probability, it outputs APSD on $G$ with worst case additive error at most $O\left(\frac{p \cdot \log^3(n/\delta)}{\varepsilon}\right)$.*

Notably, Theorem 3 offers a wide range of intriguing implications, as the definition of recursive separability captures a large class of natural graphs. Some example with implication includes

1. All $n$-vertex trees are $(1, \frac{2}{3}, \frac{2}{3} + o(1))$-recursively separable. Therefore, Theorem 3 immediately recovers the error bound for privately computing APSD on trees in [42] (though with an extra logarithmic factor).

Table 1: Current error bounds for APSD under $(\varepsilon, \delta)$-differential privacy with bounded edge weights.

| | Sealfon [42] | Chen et al. [12] | Ours |
|---|---|---|---|
| **planar graph** | $\widetilde{O}_\delta\left(\sqrt{\frac{nW}{\varepsilon}}\right)$ | $\widetilde{O}_\delta\left(\frac{W^\alpha n^{\alpha+o(1)}}{\varepsilon^{1-\alpha}} + \frac{n^{\alpha+o(1)}}{\varepsilon}\right)$ | $\widetilde{O}_\delta\left(\frac{(nW)^{1/3}}{\varepsilon^{2/3}}\right)$ |
| **grid graph** | $\widetilde{O}_\delta\left(\frac{n^{1/3}}{\varepsilon} + W \cdot n^{1/3}\right)$ | (same as above) | $\widetilde{O}_\delta\left(\frac{n^{1/4}\sqrt{W}}{\varepsilon^{1/2}}\right)$ |

2. Any graph with tree-width $p$ is $(p+1, \frac{2}{3}, \frac{2}{3} + o(1))$-recursively separable (which follows as an immediate corollary of Robertson and Seymour [41]). Therefore, Theorem 3 improves the private APSD result for bounded tree-width graphs in [23] from $\widetilde{O}(p^2/\varepsilon)$ to $\widetilde{O}(p/\varepsilon)^5$.

Leveraging the well-known separation theorem for planar graphs [28, 38], we also conclude that all planar graphs are $(O(\sqrt{n}), \frac{2}{3}, \frac{2}{3} + o(1))$-recursively separable. Though applying Theorem 3 does not immediately yield any non-trivial improvement for planar graphs, we demonstrate that with a slight modification to the recursive framework of the algorithm in Theorem 3, we can achieve an improvement not only on planar graphs but also on $K_h$-minor-free graphs. Recall that a graph is $K_h$-minor-free if the clique of size $h$, denoted by $K_h$, is not a minor of it. If a graph is $K_{h+1}$-minor-free but contains a $K_h$ minor, then $h$ is also known as the *Hadwiger number* [31].

**Theorem 4.** *Fix privacy budgets $0 < \varepsilon, \delta < 1$, and integers $W, h \geq 1$. For any weighted $K_h$-minor-free graph $G = ([n], E, w)$ with $\|w\|_\infty \leq W$, there exists a $(\varepsilon, \delta)$-differentially private algorithm such that with high probability, it outputs APSD on $G$ with worst case additive error at most $\widetilde{O}_\delta\left(h \cdot \frac{(nW)^{1/3}}{\varepsilon^{2/3}}\right)$.*

By Kuratowski's Theorem [37], planar graphs are $K_5$-minor-free. This gives the following corollary:

**Corollary 5.** *Fix privacy budgets $0 < \varepsilon, \delta < 1$, and integer $W \geq 1$. For any planar graph $G = ([n], E, w)$ with $\|w\|_\infty \leq W$, there exists a $(\varepsilon, \delta)$-differentially private algorithm such that with high probability, it outputs APSD on $G$ with worst case additive error at most $\widetilde{O}_\delta\left(\frac{(nW)^{1/3}}{\varepsilon^{2/3}}\right)$.*

Building on the same algorithmic framework, we show further improvement for a specific subclass of planar graphs: grid graphs.

**Theorem 6.** *Fix privacy budgets $0 < \varepsilon, \delta < 1$, and integers $a, W \geq 1$. For any $a \times n/a$ grid graph $G = ([a \times n/a], E, w)$ with $\|w\|_\infty \leq W$, there exists a $(\varepsilon, \delta)$-differentially private algorithm such that with high probability, it outputs APSD on $G$ with worst case additive error at most $\widetilde{O}_\delta\left(\frac{(nW^2)^{1/4}}{\varepsilon^{1/2}}\right)$.*

In Table 1, we provide a brief overview of the existing results on differentially private all-pairs shortest distances approximation when assuming an upper bound $W$ on edge weights (in this table, $\alpha = \sqrt{2} - 1 \approx 0.4143$). This assumption also appears in Sealfon [42] and Chen et al. [12].

## 1.2 Technical Overview

Here, we introduce the technical ingredients that underpin our results on the private APSD approximation and discuss the key concepts behind our analysis of privacy and utility guarantees.

### 1.2.1 Recursively computing APSD by separators

One of the most basic approaches to computing all-pair shortest distances on a graph is to add noise to the weight of each edge in $E$. This results in the error of a path being roughly proportional to the number of hops on the path, which leads to $\widetilde{O}(n/\varepsilon)$ error [42]. Using basic probability theory, it can be shown that with high probability, uniformly sampling $O(\sqrt{n} \log n)$ vertices from the vertex set of a connected graph forms a $k$-covering set (see Definition 10) of the original graph, where $k = O(\sqrt{n})$. By privately computing all-pair shortest distances within the $O(\sqrt{n})$ vertices in the covering set, the error in private APSD can be further reduced to $\widetilde{O}(\sqrt{n}/\varepsilon)$ [12, 26] since in each path, the covering set is guaranteed to be encountered after passing through the $O(\sqrt{n})$ vertices.

---

[5]This was also suggested as an open problem by Sealfon in personal communication.

However, the above methods do not make use of any combinatorial properties or topological structure of the graph. To take advantage of this, we observe that some specific classes of graphs, including trees or planar graphs, exhibit good separability, allowing us to use the divide-and-conquer method to recursively compute the shortest path. Specifically, for a graph $G = (V, E, w)$, and $x, y \in V$, let $d_G(x, y)$ be the distance between $x, y$ in $G$. Then, if $G$ has a separator $S \subseteq V$ that separates the graph into two subgraphs $G_1$ and $G_2$, we observe that for any pair $x, y \in V$ that are not in $S$, their distance can be written as

$$d_G(x, y) = \min_{a, b \in S} \{d_{G_1}(x, a) + d_G(a, b) + d_{G_2}(b, y)\}$$

if $x \in G_1$ and $y \in G_2$. Then, the error accumulated in each recursion corresponds to the error in privately computing the distances in $G$[6] between each vertex *inside* $S$. Intuitively, the recursion step can be illustrated by a binary tree of $O(\log |V|)$ levels if the input graph can be recursively divided by a series of well-balanced separators. However, this does *not* directly give the $\widetilde{O}(p/\varepsilon)$ error if the input graph is $(p, O(1), O(1))$-recursively separable, since the error accumulates across all leaf vertices in the corresponding recursion tree. To address this issue, we employ a *pruning* trick that directly establishes relationships between adjacent levels of separators, preventing errors from accumulating within the same level. In particular, we add shortcuts between adjacent levels of separators to form a series of complete bipartite graphs to control the error.

In summary, the key distinction between our technique and those employed in other works that improves private APSD [12, 26] lies in our utilization of the graph's topology to construct *highly structured* shortcuts. In contrast, the shortcuts constructed in previous works are oblivious to any combinatorial properties of the graph.

### 1.2.2 Approximating distances within a separator using covering sets

In Section 1.2.1, we briefly introduced how to construct an algorithmic framework such that the error in estimating private APSD primarily depends on the size of the separators for recursively separable graphs. Recall that in our recursive framework, we need to construct two types of shortcuts: those within the separators and those between adjacent separators. Therefore, by examining the separability of certain classes of graphs, the overall error for private APSD can be further reduced by optimizing the error incurred when estimating the distances of vertices inside or between separators. In this section, we focus solely on illustrating the idea for reducing the error in estimating the APSD (i.e., the lengths of the shortcuts) inside a separator. The idea for reducing the error in estimating the lengths of shortcuts between two adjacent separators is similar.

Specifically, suppose that the edge weights are upper-bounded by some constant $W \geq 1$. Then, we observe that finding a $k$-*covering set* (Definition 10) of the separator $S$ and computing the all-pair distances for all vertices inside the $k$-covering set provides an approximation of all-pair distances within $S$, with an extra additive error at most $kW$. The advantage is that by computing APSD only for the $k$-covering set, the number of compositions required to preserve privacy is significantly reduced when $k \ll |S|$. This approach is similar to the one by Sealfon [42] for improving the error in estimating APSD for grid graphs. However, without our recursive framework, they use this trick by finding a $k$-covering set for the whole graph instead of only for separators, which limits its ability to achieve further improvements. In our recursive framework, we observe that a grid graph is separable with a series of *connected* separators of size at most $O(\sqrt{n})$. Consequently, one can show that each separator has a $(n^{1/4})$-covering set of size at most $O(n^{1/4})$, which finally leads to an $\widetilde{O}(n^{1/4})$ error (for constant $W$) for grid graphs by the advanced composition theorem.

Achieving improvements on general planar graphs is similar but somewhat more tricky, as the separators are not necessarily connected. To make progress, for a connected planar graph $G$ on $n$ vertices, we first partition the vertex set of $G$ into about $O(n/d)$ disjoint subsets, such that each subset has diameter at most $d$ (in terms of the subgraph defined by this subset of vertices). Then, we contract each subset into a super-node, merge all multi-edges, and obtain a smaller planar graph $\widetilde{G}$ with $O(n/d)$ super-nodes. By Lipton-Tarjan's separation theorem for planar graphs [38], $\widetilde{G}$ has a separator $\widetilde{S}$ of size at most $O(\sqrt{n/d})$, and thus privately computing the all-pair shortest distances

---

[6]We have to privately compute $d_G(a, b)$ for any $a, b \in S$ instead of $d_S(a, b)$ because that the shortest path between $x, y$ may repeatedly enter and exit the separator $S$.

inside $\widetilde{S}$ incurs an error of at most $\widetilde{O}(\sqrt{n/d\varepsilon^2})$. This implies an approximation of the APSD within the original separator $S$ with an error of

$$2dW + \widetilde{O}(\sqrt{n/d\varepsilon^2}) \tag{1}$$

since the diameter of each super-node is at most $d$. By choosing an appropriate value for $d$, we obtain an $\widetilde{O}((nW)^{1/3})$ error for estimating APSD inside separators of a planar graph, and thus the same error with some extra logarithmic factors for estimating the APSD in the entire graph. Finally, this approach can also be identically applied to $K_h$-minor-free graphs (see Lemma 31). Note that the dependence on the maximum edge weight $W$ in eq.(1) is inevitable, as it ensures that the shortest paths within $S$ remain reasonably close in length to the global (true) shortest paths.

## 2 Private APSD Approximation For Recursively Separable Graph

In this section, we study answering all-pair shortest distances on *recursively separable* graphs with differential privacy. Let $G$ be a graph with non-negative edge weights, and let $x, y$ be two vertices in $G$. we use $d_G(x, y)$ to denote the (shortest) distance between $x$ and $y$ in $G$. Specifically, we establish the following result:

**Theorem 7** (Restatement of Theorem 3). *Fix any $0 < \varepsilon, \delta < 1$, and $n, p \in \mathbb{N}$. For any graph $G = ([n], E, w)$ that is $(p, q, q')$-recursively separable (Definition 1) for some constants $\frac{1}{2} \leq q \leq q' < 1$, there is an $(\varepsilon, \delta)$-differentially private algorithm such that, with high probability, it outputs estimations of shortest distances $\{\widehat{d}_G(x, y)\}_{x,y \in [n]}$ satisfying*

$$|\widehat{d}_G(s, t) - d_G(s, t)| \leq O\left(\frac{p \cdot \log^2 n \log(n/\delta)}{\varepsilon}\right).$$

**Proof outline of Theorem 7.** We provide our algorithms for privately computing APSD in recursively separable graphs in Section 2.1. Because of the space limit, we defer the privacy guarantee (Theorem 19) of our algorithms and its analysis in Appendix B.1. The utility guarantee (Theorem 20) with a detailed analysis is given in Appendix B.2. Combining Theorem 19 and Theorem 20 concludes the proof of Theorem 7.

**The corresponding result for pure-DP.** Our framework can be easily adapted to pure differential privacy by (1) replacing Gaussian noise with Laplace noise in the shortcuts, and (2) employing basic composition instead of advanced composition. Using the Laplace mechanism and the tail bound for Laplace noise, one can directly derive the following theorem from our framework. The proof is therefore omitted.

**Theorem 8.** *Fix a $p \in \mathbb{N}$. Let $G = (V, E, w)$ be a $(q, p, p')$-recursively separable graph for some constant $\frac{1}{2} \leq q \leq q' < 1$. Then, there is a $(\varepsilon, 0)$-differentially private algorithm on estimating APSD such that with probability at least $1 - \gamma$,*

$$|\widehat{d}(s, t) - d(s, t)| \leq O\left(\frac{p^2 \cdot \log^2 n (\log n + \log(1/\gamma))}{\varepsilon}\right).$$

### 2.1 The Algorithm

Our algorithm is built upon a sequence of decompositions applied to the input graph, with the resulting process being traceable through a binary tree.

**The Construction of the Binary Tree.** Given an unweighted and undirected graph $G = ([n], E)$ that is $(p, q, q')$-recursively separable, we consider a deterministic and recursive procedure that finally separates the graphs into $O(n)$ pieces of subgraphs of constant size (we note that the vertex set in these pieces may have intersection). We will see later that the bounded number of pieces is ensured by the property of recursive separability. More specifically, in the first epoch, since $G$ is $(p, q, q')$-recursively separable, then there exists an $S \subseteq [n]$ such that removing vertices in $S$ results in two subgraphs $G'_0 = (V'_0, E(V'_0)), G'_1 = (V'_1, E(V'_1))$ that are not connected, and $\max\{|V'_0|, |V'_1|\} \leq qn$. We let

$$G_0 = (V'_0 \cup S, E(V'_0 \cup S) \backslash E(S)) \quad \text{and} \quad G_1 = (V'_1 \cup S, E(V'_1 \cup S) \backslash E(S)). \tag{2}$$

That is, we union $S$ into $G'_1$ (or $G'_2$) together with all edges incident between $S$ and $G'_1$ (or $G'_2$). From the recursive separability of $G$, we also have that

$$\max\{|V_0|, |V_1|\} \leq qn + |S| \leq q'n,$$

where $V_0$ (or $V_1$) is the set of vertices of $G_0$ (or $G_1$).

In the recursive subroutine, for graph $G_b$ where $b$ is a binary string of length that depends on the current depth of recurrence, there must exist a separator $S_b$ that separates $G_b$ into $G'_{b \circ 1}$ and $G'_{b \circ 0}$. Similarly as in eq. (2), we let

$$G_{b \circ 1} = (V'_{b \circ 1} \cup S_b, E(V'_{b \circ 1} \cup S_b) \backslash E(S_b)) \quad \text{and} \quad G_{b \circ 0} = (V'_{b \circ 0} \cup S_b, E(V'_{b \circ 0} \cup S_b) \backslash E(S_b)).$$

The recursive procedure terminates when $G_b$ has constant size $c$. Clearly, the depth of the recursion is $\log_{1/q'} \frac{n}{c} = \frac{\log(n/c)}{\log(1/q')} = O(\log n)$ for any constant $0 < q' < 1$. Further, these graphs construct a binary tree $\mathcal{T}$ with $O(\log n)$ levels, where the root of $\mathcal{T}$ is $G$, and the nodes in the $i$-th level is just the all $G_b$'s with $|b| = i$, and each $G_b$ in the leaves of $\mathcal{T}$ has size at most $c$. In each non-leaf node of $\mathcal{T}$, we associate it with a label $(G_b, S_b)$ to denote the subgraph it represents and the separator used to further split $G_b$. We note that for the root, $b = \varnothing$.

**The Algorithm.** With the construction of the binary tree described above, we are ready to present our algorithm. For keeping the presentation modular, we present it in form of three algorithms: Algorithm 1 defines how we add shortcuts, Algorithm 3 describes the recursive procedure that is used by Algorithm 2 to compute all-pair-shortest distances.

In Algorithm 1, we first add highly structured shortcuts based on the decomposition described above to generate a private synthetic graph. In particular, there are two types of shortcuts: (1) between all pairs of vertices within each separator that is determined by the decomposition procedure, forming a complete graph, and (2) between vertices in every pair of adjacent separators, forming a complete bipartite graph.

Then, as a post-processing stage, in Algorithm 2 we compute the all-pair shortest path distances privately for a weighted graph by calling the recursive procedure for each pair (Algorithm 3). Here, for any $s, t \in V$, we use $d_b(s, t)$ to denote the value of the *local* shortest distance between $s$ and $t$ in the subgraph $G_b$. If either $s$ or $t$ does not appear in $V_b$, then we set $d_b(s, t) = \infty$. In our algorithms, for any $x, y \in [n]$ and $b \in \{0, 1\}^*$, we define $\texttt{IsShortcut}(x, y, b) = \textit{True}$ if a noisy edge (i.e., a shortcut) is added between $x, y$ within the separator of $G_b$, or between the separator of $G_b$ and that of its precursor graph.

## 3 Improved Results for $K_h$-Minor-Free Graphs

In this section, we utilize the algorithmic framework proposed in Section 2 to present our improved results on privately answering APSD for some special classes of graphs. In particular, assuming bounded weights, where the maximum weight of edges is denoted as $W$, and by some specific separation theorems for special graphs, we can use the general framework (along with a sub-sampling trick inside the separator) to achieve a purely additive error of $\sim (nW)^{1/3}$ when privately approximating APSD in $K_h$-minor-free graphs for any constant $h$. Additionally, as a special subclass of $K_5$-minor free graphs, we are able to leverage some unique structural properties of grid graphs to further reduce the additive error to approximately $\sim n^{1/4}\sqrt{W}$.

**Theorem 9.** *Fix any $0 < \varepsilon, \delta < 1$ and $n \in \mathbb{N}$. There exists an $(\varepsilon, \delta)$-differentially private algorithm for estimating all-pairs shortest path distances in $G$ such that, with high probability, the worst-case error is bounded by $O\left(\frac{n^{1/4}\sqrt{W} \cdot \log^2 n}{\varepsilon^{1/2}} \log\left(\frac{\log n}{\delta}\right)\right)$ for any grid graph $G = ([a \times b], E, w)$ where $ab = n$. Further, the error is bounded by $O\left(h \cdot \frac{(nW)^{1/3} \cdot \log^2 n}{\varepsilon^{2/3}} \log\left(\frac{\log n}{\delta}\right)\right)$ with high probability, for any $K_h$-minor-free graph $G = ([n], E, w)$.*

**Proof outline of Theorem 9.** We first present here the modifications to the recursive framework described in Section 2 that are required to derive Theorem 9, while deferring the detailed analysis of it to Appendix C. Specifically, instead of constructing the all-pair shortcut inside or between separators as in Algorithm 1, we find a $k$-covering set of each separator and only add shortcuts inside or between such covering sets to reduce the number of compositions needed to preserve privacy.

---
**Algorithm 1:** Constructing private shortcuts.
---
**Input:** Graph $G = (V, E, w)$, private parameter $\varepsilon, \delta$.

1. Recursively construct a binary tree $\mathcal{T}$ as described in this section.

2. Set $h = \log_{1/q'}(n/c)$, $\varepsilon' = \varepsilon/\sqrt{4h \log(1/\delta')}$, and $\delta' = \delta/(4h)$.

3. Let $\sigma = p\sqrt{2\log(1.25/\delta')}/\varepsilon'$.

4. **for** *non-leaf node* $(G_b, S_b) \in V(\mathcal{T})$ **do**
    **for** $x, y \in S_b$ *such that* $x \neq y$ **do**
        IsShortcut$(x, y, b) =$ *True*.
        Let $\widehat{d}_b(x, y) = d_b(x, y) + \mathcal{N}(0, \sigma^2)$.
    **end**
    **if** $b \neq \varnothing$ **then**
        Let $b'$ be the binary string that removes the last bit in $b$.
        **for** $(x, y) \in S_{b'} \times S_b$ **do**
            **if** $x, y \notin S_{b'} \cap S_b$ **then**
                IsShortcut$(x, y, b') =$ *True*.
                Let $\widehat{d}_b(x, y) = d_b(x, y) + \mathcal{N}(0, \sigma^2)$.
            **end**
        **end**
    **end**
**end**

5. **for** *leaf node* $(G_b, \text{-}) \in V(\mathcal{T})$ **do**
    **for** $x, y \in V_b$ *such that* $x \neq y$ **do**
        IsShortcut$(x, y, b) =$ *True*.
        Let $\widehat{d}_b(x, y) = d_b(x, y) + \mathcal{N}\left(0, \frac{2c^2 \log(1.25/\delta')}{(\varepsilon')^2}\right)$.
    **end**
**end**

**Output :** The binary tree $\mathcal{T}$, and $\widehat{d}_b(u, v)$ for all $u, v \in V$ and $b \in \{0, 1\}^*$ such that
        IsShortcut$(x, y, b) =$ *True*.

---

---
**Algorithm 2:** Differentially private all-pair-shortest-path approximation
---
**Input:** Graph $G = (V, E, w)$, private parameter $\varepsilon, \delta$.

1. Run Algorithm 1 on $G$ and $\varepsilon, \delta$, obtain the labeled binary tree $\mathcal{T}$.

2. **for** $s, t \in V$ *such that* $s \neq t$ **do**
    $\widehat{d}(x, y) \leftarrow$ Recursive-APSD$(\mathcal{T}, G, (s, t), 0)$.            // Algorithm 3
**end**

**Output :** estimated distances $\widehat{d}(x, y)$ on $G$.

---

**Definition 10** (*k*-covering)**.** Given a graph $G = (V, E)$, a subset $Z \subseteq V$ is a *k*-covering of $V$ if for every vertex $a \in V$, there is a vertex $b \in Z$ such that the hop distance between $a$ and $b$ is at most $k$.

**The construction of the binary tree for $K_h$-minor-free graphs** is similar to that for general graphs, and that a $K_h$-minor-free graph $G$ with $n$ vertices are guaranteed to always have a (not necessarily connected) separator $S$ of size at most $O(\sqrt{h^3 n})$, such that removing $S$ results in two disjoint subgraphs $G_0'$ and $G_1'$ (see Lemma 27). Both subgraphs are clearly also $K_h$-minor-free, and are of size at most $2n/3$. This process can be repeated recursively, splitting each subgraph until every part has constant size. For any grid graph on $n$ vertices of shape $a \times b$, we can split the graph into two sub-grid graphs of (almost) equal size using a separator of size at most $O(\sqrt{n})$, and that the separator is *connected*. Hence we have the lemma below.

**Lemma 11.** *Fix an integer $h \geq 1$, any $K_h$-minor-free graph $G = ([n], E)$ is $(c\sqrt{h^3 n}, 2/3, 2/3 + ch^{3/2}/\sqrt{n})$-recursively separable for some constant $c$. In particular, planar graphs (as well as grid graphs $G_{grid} = ([a \times b], E)$ with $ab = n$) are $(c'\sqrt{n}, 2/3, 2/3 + o(1))$-recursively separable for some constant $c'$.*

**Algorithm 3:** Recursive-APSD$(\mathcal{T}, G_b, (s,t), k)$

---

**Input:** A binary tree constructed out of some graph $G = (V, E, w)$, a graph $G_b$, a pair of vertices $s, t \in V_b$, integer $k \in \mathbb{N}$.

1. **if** $\mathit{IsShortcut}(s, t, b) = \mathsf{True}$ **then**
  Halt and directly output $\widehat{d}_b(s,t)$ computed by Algorithm 1.
**end**

2. **if** $k = 0$ **then**
  **if** *both* $s, t \in V_{b \circ a}$ *for* $a \in \{0, 1\}$ **then**
    **for** $z \in S_b$ **do**
      $\widehat{d}_{b \circ a}(s, z) \leftarrow$ Recursive-APSD$(\mathcal{T}, G_{b \circ a}, (s, z), k+1)$.
      $\widehat{d}_{b \circ a}(t, z) \leftarrow$ Recursive-APSD$(\mathcal{T}, G_{b \circ a}, (t, z), k+1)$.
    **end**
    $\widehat{d}_{b \circ a}(s, t) \leftarrow$ Recursive-APSD$(\mathcal{T}, G_{b \circ a}, (s, t), 0)$.
    $\widehat{d}_b(s, t) \leftarrow \min\{\widehat{d}_{b \circ a}(s, t), \min_{x, y \in S_b} \widehat{d}_{b \circ a}(s, x) + \widehat{d}_b(x, y) + \widehat{d}_{b \circ a}(y, t)\}$
  **end**
  **if** $s \in V_{b \circ a}$ *and* $t \in V_{b \circ \bar{a}}$, **then**
    **for** $z \in S_b$ **do**
      $\widehat{d}_{b \circ a}(s, z) \leftarrow$ Recursive-APSD$(\mathcal{T}, G_{b \circ a}, (s, z), k+1)$.
      $\widehat{d}_{b \circ \bar{a}}(t, z) \leftarrow$ Recursive-APSD$(\mathcal{T}, G_{b \circ \bar{a}}, (t, z), k+1)$.
    **end**
    $\widehat{d}_b(s, t) \leftarrow \min_{x, y \in S_b} \widehat{d}_{b \circ a}(s, x) + \widehat{d}_b(x, y) + \widehat{d}_{b \circ \bar{a}}(y, t)$
  **end**
**end**

3. **if** $k > 0$ **then**
  Let $b'$ be the binary string that removes the last bit in $b$.
  **if** $s, t \notin S_{b'}$ **then**
    Halt and output `FAIL`.
  **end**
  WLOG let $t \in S_{b'}$ (if not we just switch $s$ and $t$).
  **for** $z \in S_b$ **do**
    $\widehat{d}_{b \circ a}(s, z) \leftarrow$ Recursive-APSD$(\mathcal{T}, G_{b \circ a}, (s, z), k+1)$.
  **end**
  **if** *both* $s, t \in V_{b \circ a}$ *for* $a \in \{0, 1\}$ **then**
    $\widehat{d}_{b \circ a}(s, t) \leftarrow$ Recursive-APSD$(\mathcal{T}, G_{b \circ a}, (s, t), 0)$.
    Let $\widehat{d}_b(s, t) \leftarrow \min\{\widehat{d}_{b \circ a}(s, t), \min_{x \in S_b} \widehat{d}_{b \circ a}(s, x) + \widehat{d}_b(x, t)\}$.
  **end**
  **if** $s \in V_{b \circ a}$ *and* $t \in V_{b \circ \bar{a}}$, **then**
    $\widehat{d}_b(s, t) \leftarrow \min_{x \in S_b} \widehat{d}_{b \circ a}(s, x) + \widehat{d}_b(x, t)$.
  **end**
**end**

**Output:** estimated local distance $\widehat{d}_b(s, t)$.

---

As we introduced earlier, with this decomposition, the only modification required is in Algorithm 1. In Algorithm 1, we construct private shortcuts by connecting all pairs of vertices in separator. We now modify Algorithm 1 into Algorithm 4 as follows: for each separator, we begin by finding a $k$-covering for it. Then, we construct private shortcuts by connecting all pairs of vertices *within* the $k$-covering sets. We provide the complete pseudocode for Algorithm 4 in Appendix D, and here we only highlight the differences. The modified step 4 of Algorithm 1 is as follows.



**Step 4 of Algorithm 4 (modified step 4 of Algorithm 1):**
**for** *non-leaf node* $(G_b, S_b) \in V(\mathcal{T})$ **do**

    Find a $k$-covering set of $S_b$, and let it be $S_b^k$.

    **for** $x, y \in S_b^k$ *such that* $x \neq y$ **do**

        $\texttt{IsShortcut}(x, y, b) = \textit{True}$. Let $\widehat{d}_b(x, y) = d_b(x, y) + \mathcal{N}(0, \sigma^2)$.

    **end**

    **if** $b \neq \varnothing$ **then**

        Let $b'$ be the binary string that removes the last bit in $b$.

        Find a $k$-covering set of $S_{b'}$, and let it be $S_{b'}^k$.

        **for** $(x, y) \in S_{b'}^k \times S_b^k$ **do**

            **if** $x, y \notin S_{b'}^k \cap S_b^k$ **then**

                $\texttt{IsShortcut}(x, y, b') = \textit{True}$. Let $\widehat{d}_b(x, y) = d_b(x, y) + \mathcal{N}(0, \sigma^2)$.

            **end**

        **end**

    **end**

**end**



We also modify Step 3 of Algorithm 1 to revise the number of compositions needed for privacy from $O(p^2)$ to $O(f^2(p, k))$, as follows:

**Step 3 of Algorithm 4**: Let $\sigma = f(p, k)\sqrt{2\log(1.25/\delta')}/\varepsilon'$, where $f(p, k)$ is the upper bound of the size of the $k$-covering set for the separator of size $p$.

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

# A   Notations and Preliminaries

Throughout this paper, we work on weighted graphs $G = (V, E, w)$ where $w \in \mathbb{R}_+^E$ encodes edge weights. Let $a, b$ be two binary strings, we use $a \circ b$ to denote the binary string obtained concatenating $a$ and $b$. When edge weights are not of concern, for a graph $G = (V, E)$ and $S \subseteq V$, we use $G_S = (S, E(S))$ to denote the subgraph induced by $S$. For any $h \geq 1$, we use $K_h$ to denote a clique of size $h$. A graph $H$ is a minor of a graph $G$ if a copy of $H$ can be obtained from $G$ via repeated edge deletion or edge contraction, and we say a graph $G$ is $K_h$-*minor-free* if $G$ does not have $K_h$ as its minor. It is well-known that all planar graphs are $K_5$-minor-free:

**Lemma 12** (The Kuratowski's reduction theorem [37, 45])**.** *A graph $G$ is planar if and only if the complete graph $K_5$ and the complete bipartite graph $K_{3,3}$ are not minors of $G$.*

Here, we provide the necessary background on differential privacy to facilitate understanding of this paper. A key feature of differential privacy algorithms is that they preserve privacy under post-processing. That is to say, without any auxiliary information about the dataset, any analyst cannot compute a function that makes the output less private.

**Lemma 13** (Post processing [21])**.** *Let $\mathcal{A} : \mathcal{X} \to \mathcal{R}$ be a $(\varepsilon, \delta)$-differentially private algorithm. Let $f : \mathcal{R} \to \mathcal{R}'$ be any function, then $f \circ \mathcal{A}$ is also $(\varepsilon, \delta)$-differentially private.*

Sometimes we need to repeatedly use differentially private mechanisms on the same dataset, and obtain a series of outputs.

**Lemma 14** (Basic composition [18])**.** *let $D$ be a dataset in $\mathcal{X}$ and $\mathcal{A}_1, \mathcal{A}_2, \cdots, \mathcal{A}_k$ be $k$ algorithms where $\mathcal{A}_i$ (for $i \in [k]$) preserves $(\varepsilon_i, \delta_i)$ differential privacy, then the composed algorithm $\mathcal{A}(D) = (\mathcal{A}_1(D), \cdots, \mathcal{A}_2(D))$ preserves $(\sum_{i \in [k]} \varepsilon_i, \sum_{i \in [k]} \delta_i)$-differential privacy.*

**Lemma 15** (Advanced composition [22])**.** *For parameters $\varepsilon > 0$ and $\delta, \delta' \in [0, 1]$, the composition of $k$ $(\varepsilon, \delta)$-differentially private algorithms is a $(\varepsilon', k\delta + \delta')$ differentially private algorithm, where*

$$\varepsilon' = \sqrt{2k \log(1/\delta')} \cdot \varepsilon + k\varepsilon(e^\varepsilon - 1).$$

The Laplace mechanism is one of the most basic mechanisms to preserve differential privacy for numeric queries, which calibrates a random noise from the Laplace distribution (or double exponential distribution) according to the $\ell_1$ sensitivity of the function.

**Lemma 16.** *(Laplace mechanism) Suppose $f : \mathcal{X} \to \mathbb{R}^k$ is a query function with $\ell_1$ sensitivity $\mathsf{sens}_1(f) \leq \Delta_1$. Then the mechanism*

$$\mathcal{M}(D) = f(D) + (Z_1, \cdots, Z_k)^\top$$

*is $(\varepsilon, 0)$-differentially private, where $Z_1, \cdots, Z_k$ are i.i.d random variables drawn from $\mathtt{Lap}(\Delta_1/\varepsilon)$.*

Adding Gaussian noise based on the $\ell_2$ sensitivity guarantees approximate differential privacy.

**Lemma 17.** *(Gaussian mechanism) Suppose $f : \mathcal{X} \to \mathbb{R}^k$ is a query function with $\ell_2$ sensitivity $\mathsf{sens}_1(f) \leq \Delta_2$. Then the mechanism*

$$\mathcal{M}(D) = f(D) + (Z_1, \cdots, Z_k)^\top$$

*is $(\varepsilon, \delta)$-differentially private, where $Z_1, \cdots, Z_k$ are i.i.d random variables drawn from $\mathcal{N}\left(0, \frac{(\Delta_2)^2 \cdot 2\ln(1.25/\delta)}{\varepsilon}\right)$.*

# B   Proof of Theorem 7

## B.1   Privacy Analysis

Here, we analyze the privacy guarantee of our algorithm. We observe that constructing the binary tree $\mathcal{T}$ does not compromise privacy, as it only requires the topology, not the edge weights, as input. Consequently, the recursive procedure (Algorithm 3) and Algorithm 2 involving the tree $\mathcal{T}$ are simply post-processing steps of Algorithm 1. Therefore, we focus our privacy analysis solely on Algorithm 1. We use the following lemma to bound the sensitivity of the binary tree that traces the decomposition:

**Lemma 18.** *Let $\mathcal{T}$ be the labeled tree constructed from $G$. For any vertex $u, v$ with an edge $e = \{u, v\}$ between them, $e$ appears in at most $h = \log_{1/q'}(n/c)$ nodes in $\mathcal{T}$.*

*Proof.* Suppose $e$ is in the edge set of some internal node $G_b$. Recall that in the division of $G_b$, the edge $e$ will be removed if and only if $u, v \in S_b$. Then $e$ only exists in any $G_{b'}$ such that $b'$ is the prefix of $b$. Since $|b| \leq h$, the lemma holds.

On the other hand, if $e$ is in the edge set of some leaf node $G_b$. Then the recursive construction terminates on $G_b$, and $e$ only exists in any $G_{b'}$ such that $b'$ is the prefix of $b$, again the number of such nodes is at most $h$. This completes the proof of Lemma 18. $\qquad\square$

We are now ready to prove the following theorem on the privacy guarantee, utilizing the advanced composition lemma (Lemma 15).

**Theorem 19.** *Algorithm 2 preserves $(\varepsilon, \delta)$-differential privacy.*

*Proof.* Suppose for a pair of neighboring graphs $G$ and $G'$, they have a difference in the edge weight by $1$ of $e = \{u, v\}$. Let $G_b$ be the subgraph of $G$ that contains $e$ in its edge set.

1. If $G_b$ is an internal node of $\mathcal{T}$. let $d_{S_b} = \{d_b(x, y) : x \neq y \wedge (x, y) \in S_b\}$ be the vector of distances between vertices in the separator of $G_b$. Since $|d_b(x, y) - d_b'(x, y)| \leq 1$ for any distinct $x, y \in S_b$, then $\|d_{S_b}\|_2 \leq p$ as $|S_b \times S_b| \leq p^2$. Then, by the Gaussian mechanism, outputting $\widehat{d}_b(x, y)$ for all distinct $x, y \in S_b$ preserves $(\varepsilon', \delta')$-differential privacy. Similarly, outputting $\widehat{d}_b(x, y)$ for all $(x, y) \in S_b \times S_{b'}$ where $x, y \notin S_{b'} \cap S_b$ also preserves $(\varepsilon', \delta')$-differential privacy, as $|S_b \times S_{b'}| \leq p^2$.

2. If $G_b$ is a leaf node of $\mathcal{T}$. Since $V_b \leq c$, then again be the Gaussian mechanism, outputting the all-pair-shortest distance $\widehat{d}_b(\cdot, \cdot)$ in $G_b$ preserves $(\varepsilon', \delta')$-differential privacy.

Combining the above argument, Lemma 18 and the advanced composition, we have that Algorithm 1 preserves $(\varepsilon, \delta)$-DP, as well as Algorithm 2. This completes the proof of Theorem 19. $\qquad\square$

### B.2 Utility Analysis

Here, we present the utility guarantee of Algorithm 2, which privately computes the all-pair shortest distances for the input graph by repeatedly invokes Algorithm 3 for each pair.

**Theorem 20.** *Fix any $0 < c \leq n$ and any $0 < \gamma < 1$. Let $G = (V, E, w)$ be a $(p, q, q')$-recursively separable graph for some $p \in \mathbb{N}$ and $\frac{1}{2} \leq q \leq q' < 1$. Then with probability at least $1 - \gamma$, we have that for any $s, t \in V$,*

$$|\widehat{d}(s, t) - d(s, t)| \leq O\left(\frac{(hp + c) \cdot \log(h/\delta) \cdot \sqrt{h^2 + h\log(\max\{p, c\})} + h\log(1/\gamma)}{\varepsilon}\right),$$

*where $h = \log_{1/q'}(n/c)$. That is, for any constant $c$ and $q'$, we have with high probability,*

$$|\widehat{d}(s, t) - d(s, t)| \leq O\left(\frac{p\log^2 n \cdot \log\left(\frac{\log n}{\delta}\right)}{\varepsilon}\right).$$

To prove Theorem 20. First, we observe that, with probability 1, Algorithm 3 terminates normally without any abnormal termination.

**Fact 21.** *For any pair of vertices $s, t \in V$, Algorithm 3 does not output "FAIL".*

*Proof.* The only case when Algorithm 3 outputs "FAIL" is that the parameter $k \geq 1$ and both $s, t$ are not in the separator of the predecessor graph of $G_b$. However, it is not possible because when Algorithm 3 is called with $k \geq 1$, at least one of the vertex in $s, t$ is from the separator $S_{b'}$ where $G_b = G_{b' \circ 0}$ or $G_b = G_{b' \circ 1}$. $\qquad\square$

The following lemma is used for building the correctness of the recursion.

**Lemma 22.** *Fix any $s, t \in V_b$. Without the lose of generality, we assume either both $s, t \in V_{b \circ 0}$ or $s \in V_{b \circ 0}$ and $t \in V_{b \circ 1}$ (otherwise we just switch $s$ and $t$, and the proof for both $s, t \in V_{b \circ 1}$ is symmetric). We have*

$$d_b(s, t) = \min \left\{ d_{b \circ 0}(s, t), d_{b \circ 1}(s, t), \min_{x, y \in S_b} d_{b \circ 0}(s, x) + d_b(x, y) + d_{b \circ 1}(y, t) \right\}.$$

*Proof.* We discuss by difference cases:

**1. When at least one of $s, t$ is in $S_b$.** By the construction of $G_{b \circ a}$ such that

$$G_{b \circ a} = (V'_{b \circ a} \cup S, E(V_{b \circ a} \cup S_b) \backslash E(S_b)),$$

we have $d_b(s, t) \geq d_{b \circ 0}(s, t)$ and $d_b(s, t) \geq d_{b \circ 1}(s, t)$. Similarly, we have that

$$\min_{x, y \in S_b} d_{b \circ 0}(s, x) + d_b(x, y) + d_{b \circ 1}(y, t) \geq \min_{x, y \in S_b} d_b(s, x) + d_b(x, y) + d_b(y, t) \geq d_b(s, t) \quad (3)$$

by the triangle inequality. On the other hand, if both $s, t \in S_b$, then we have

$$d_b(s, t) = d_{b \circ 0}(s, s) + d_b(s, t) + d_{b \circ 1}(t, t) \geq \min_{x, y \in S_b} d_{b \circ 0}(s, x) + d_b(x, y) + d_{b \circ 1}(y, t). \quad (4)$$

If exactly one of $s, t \in S_b$, we note that any path $(v_1 = s, v_2, \cdots, v_k = t)$ from $s$ to $t$, must satisfy that $v_i \in S_b$ for at least one $i \in [k]$ since $s \in S_b$ or $t \in S_b$. Again by the symmetry, we assume $t \in S_b$. Suppose $z$ is the first vertices in $S_b$ of any shortest path from $s$ to $t$. Then,

$$d_b(s, t) = d_{b \circ 0}(s, z) + d_b(z, t) + d_{b \circ 1}(t, t) \geq \min_{x, y \in S_b} d_{b \circ 0}(s, x) + d_b(x, y) + d_{b \circ 1}(y, t). \quad (5)$$

Combining Equation (3), Equation (4) and Equation (5) completes the proof in this case.

**2. When both $s, t \notin S_b$.** First, we assume that $s$ and $t$ are on different sides such that $s \in V_{b \circ 0}$ and $t \in V_{b \circ 1}$. In this case $d_{b \circ 0}(s, t) = d_{b \circ 1}(s, t) = \infty$. By the fact that $S_b$ is a separator, any path $(v_1 = s, v_2, \cdots, v_k = t)$ from $s$ to $t$ must also satisfy that $v_i \in S_b$ for at least one $i \in [k]$. Let $z_1, z_2$ be the first and last vertices in one of the shortest paths from $s$ to $t$ such that $z_1, z_2 \in S_b$ (we allow $z_1 = z_2$). Then it is easy to verify that $d_b(s, z_1) = d_{b \circ 0}(s, z_1)$ and $d_b(z_2, t) = d_{b \circ 1}(z_2, t)$. Thus,

$$\min_{x, y \in S_b} d_{b \circ 0}(s, x) + d_b(x, y) + d_{b \circ 1}(y, t) \leq d_{b \circ 0}(s, z_1) + d_b(z_1, z_2) + d_{b \circ 1}(z_2, t) = d_b(s, t).$$

Again by the triangle inequality, we also have Equation (3) holds, which proves Lemma 22. Now, suppose that both $s$ and $t$ are in the same side $V_{b \circ 0}$. Then the shortest path either crosses $S_b$ or not. If the path crosses $S_b$, then the previous argument suffices to prove Lemma 22. If the path does not crosses $S_b$, then we have $d_b(s, t) = \min\{d_{b \circ 0}(s, t), d_{b \circ 1}(s, t)\}$, which completes the proof. $\square$

The following lemma utilizes the shortcut between the separators of two adjacent layers (built by step 4 of Algorithm 1) for pruning.

**Lemma 23.** *Let $S_{b'}$ and $S_b$ be two adjacent separators where $b = b' \circ u$ for $u \in \{0, 1\}$. Then for any $x \in S_b$, $a \in \{0, 1\}^n$ and $y \in S_{b'}$,*

$$d_b(x, y) = \min_{z \in S_b} d_b(x, z) + d_{b \circ a}(z, y).$$

*Proof.* Let $P = (v_1 = x, v_2, \cdots, v_k = y)$ be any shortest path from $x$ to $y$, and let $w$ be the last vertex in $P$ such that $w \in S_b$. Such $w$ exists since $x \in S_b$. Then,

$$d_b(x, y) = d_b(x, w) + d_b(w, y) = d_b(x, w) + d_{b \circ a}(w, y) \geq \min_{z \in S_b} d_b(x, z) + d_{b \circ a}(z, y).$$

On the other hand,

$$d_b(x, y) \leq \min_{z \in S_b} d_b(x, z) + d_b(z, y) \leq \min_{z \in S_b} d_b(x, z) + d_{b \circ a}(z, y)$$

due to the triangle inequality and the fact that $d_b(z, y) \leq d_{b \circ a}(z, y)$. This completes the proof of Lemma 23. $\square$

Without the lose of generality, we assume the $\mathcal{T}$ is a complete binary tree of height $h$. (If one branch terminates early, we just let it continue to split with one of its branches be an empty graph, until the height is $h$.) We need to use the following observation to control the accumulation of the error:

**Fact 24.** *During the execution of Algorithm 2, for any $b \in \{0,1\}^*$ such that $|b| \leq h$ and any pair of vertices $s, t \in V_b$, Algorithm 3 with parameter $G_b$ and $(s,t)$ is invoked exactly once.*

*Proof.* We prove it by induction on the size of $b$. We claim that if Recursive-APSD$(\mathcal{T}, G_b, (s,t), k)$ where $|b| \leq h - 1$ is called for any $s, t \in G_b$ and some $k$, then both Recursive-APSD$(\mathcal{T}, G_{b\circ 0}, (s,t), k')$ and Recursive-APSD$(\mathcal{T}, G_{b\circ 1}, (s,t), k')$ will be called for some $k'$. This is clearly true since Recursive-APSD$(\mathcal{T}, G_{b\circ a}, (s,t), k+1)$ will be invoked if both $s$ and $t$ lies in $G_{b\circ a}$. Then, since Algorithm 2 invokes Algorithm 3 for any distinct $s, t \in V$, we conclude that Algorithm 3 will be invoked at least once for every $G_b$ and all-pair vertices in $G_b$. On the other hand, since Algorithm 3 with $G_b$ as parameter will only be invoked by $b'$ such that $b = b' \circ a$ for $a \in \{0,1\}$, then it will only be invoked once. □

Next, we analyze the error on each shortcut. We write $\sigma_1 = c \cdot \sqrt{2\log(1.25/\delta')}/\varepsilon'$ and $\sigma_2 = p \cdot \sqrt{2\log(1.25/\delta')}/\varepsilon'$.

**Lemma 25.** *With probability at least $1 - \gamma$, both the following holds:*

1. *For any $s, t, b$ where $|b| = h$ and that $\mathtt{IsShortcut}(s,t,b) = \mathsf{True}$,*

$$|d_b(s,t) - \widehat{d}_b(s,t)| \leq \sigma_1 \cdot \sqrt{2\left(h + 3\ln(\max\{p,c\}) + \ln\left(\frac{1}{2\gamma}\right)\right)}$$

2. *For any $s, t, b$ where $|b| < h$ and that $\mathtt{IsShortcut}(s,t,b) = \mathsf{True}$,*

$$|d_b(s,t) - \widehat{d}_b(s,t)| \leq \sigma_2 \cdot \sqrt{2\left(h + 3\ln(\max\{p,c\}) + \ln\left(\frac{1}{2\gamma}\right)\right)}.$$

*Proof.* We recall that for the Gaussian variable $X \sim \mathcal{N}(0, \sigma^2)$, $\mathbf{Pr}[|X| \geq t] \leq \exp(-t^2/2\sigma^2)$ for any $t \geq 0$. For any $s, t, b$ with $\mathtt{IsShortcut}(s,t,b) = \mathsf{True}$ and $G_b$ is a leaf node, the difference in $\widehat{d}_b(s,t)$ and $d_b(s,t)$ is a Gaussian noise with variance $\sigma_1$, and thus

$$\mathbf{Pr}[d_b(s,t) - \widehat{d}_b(s,t) \geq z] \leq \frac{\gamma}{m}$$

if we choose $z = \sigma_1\sqrt{2\log(m/(2\gamma))}$ for any $m \geq 1$ and $0 < \gamma < 1$. Also, if $G_b$ is not a leaf node, then according to Algorithm 1, we have

$$\mathbf{Pr}\left[d_b(s,t) - \widehat{d}_b(s,t) \geq \sigma_2\sqrt{2\log m/(2\gamma)}\right] \leq \frac{\gamma}{m}.$$

The number of noises added in Algorithm 1 is bounded by $m = 5 \cdot 2^h \cdot \max\{p^2, c^2\}$. Lemma 25 now follows using the union bound. □

With all the aforementioned preparations, we are now ready to prove Theorem 20 regarding the utility guarantee of Algorithm 2 by induction. We restate this theorem here.

**Theorem 26** (Restatement of Theorem 20). *Fix any $0 < c \leq n$ and any $0 < \gamma < 1$. Let $G = (V, E, w)$ be a $(p, q, q')$-recursively separable graph for some $p \in \mathbb{N}$ and $\frac{1}{2} \leq q \leq q' < 1$. Then with probability at least $1 - \gamma$, we have that for any $s, t \in V$,*

$$|\widehat{d}(s,t) - d(s,t)| \leq O\left(\frac{(hp + c) \cdot \log(h/\delta) \cdot \sqrt{h^2 + h\log(\max\{p,c\}) + h\log(1/\gamma)}}{\varepsilon}\right),$$

*where $h = \log_{1/q'}(n/c)$. That is, for any constant $c$ and $p'$, we have with high probability,*

$$|\widehat{d}(s,t) - d(s,t)| \leq O\left(\frac{p \cdot \log^2 n \log(n/\delta)}{\varepsilon}\right).$$

*Proof.* With the help of Fact 24, we prove this theorem by induction on the size of $b$. We first define

$$\texttt{err}(b) = \zeta_1 + (h - |b|)\zeta_2.$$

for any $b \in \{0,1\}^*$ and $|b| \le h$. Here, $\zeta_1 = \sigma_1 \cdot \sqrt{2(h + 3\ln(\max\{p,c\}) + \ln(1/(2\gamma)))}$ and $\zeta_2 = \sigma_2 \cdot \sqrt{2(h + 3\ln(\max\{p,c\}) + \ln(1/(2\gamma)))}$. Then, it is sufficient to just show that for any $G_b$ in the binary tree $\mathcal{T}$ and any $x, y \in V_b$,

$$|d_b(s,t) - \widehat{d}_b(s,t)| \le 2\texttt{err}(b),$$

as letting $b = \varnothing$ completes the proof. For the base case, this is true for all $|b| = h$ because in this case $G_b$ is the leaf node of $\mathcal{T}$ and thus $|d_b(s,t) - \widehat{d}_b(s,t)| \le \zeta_1 = \texttt{err}(b)$. For all $|b| < h$ and $\texttt{IsShortcut}(s,t,b) = \textit{True}$, we also have that

$$|d_b(s,t) - \widehat{d}_b(s,t)| \le \zeta_2 \le \zeta_1 + \zeta_2 \le \texttt{err}(b).$$

Suppose that for any $s, t$ and $b$ with $|b| < h$, we always have

1. If $\widehat{d}_b(s,t)$ is computed by the unique invocation of Algorithm 3 with parameter $k > 0$, then $|d_{b \circ a}(s,t) - \widehat{d}_{b \circ a}(s,t)| \le \texttt{err}(b \circ a)$ for $a \in \{0,1\}$;

2. If $\widehat{d}_b(s,t)$ is computed by the unique invocation of Algorithm 3 with parameter $k = 0$, then $|d_{b \circ a}(s,t) - \widehat{d}_{b \circ a}(s,t)| \le 2\texttt{err}(b \circ a)$ for $a \in \{0,1\}$.

Now we look at the induction case.

**Case(1)** Suppose $\widehat{d}_b(s,t)$ is computed by the unique invocation of Algorithm 3 with parameter $k > 0$ and that $\texttt{IsShortcut}(s,t,b) = \text{False}$. We only analysis the case when both $s, t \in V_{b \circ a}$ for $a \in \{0,1\}$, since the proof for the case where $s, t$ are in the different sides is identical. In this case, by Algorithm 3,

$$\widehat{d}_b(s,t) = \min\{\widehat{d}_{b \circ a}(s,t), \min_{x \in S_b} \widehat{d}_{b \circ a}(s,x) + \widehat{d}_b(x,t)\}.$$

On the other hand, from Lemma 22, we have that

$$d_b(s,t) = \min\left\{d_{b \circ a}(s,t), \min_{x,y \in S_b} d_{b \circ a}(s,x) + d_b(x,y) + d_{b \circ a}(y,t)\right\},$$

where

$$\min_{x,y \in S_b} d_{b \circ a}(s,x) + d_b(x,y) + d_{b \circ a}(y,t) = \min_{x \in S_b}\left(d_{b \circ a}(s,x) + \min_{y \in S_b} d_b(x,y) + d_{b \circ a}(y,t)\right) \tag{6}$$
$$= \min_{x \in S_b} d_{b \circ a}(s,x) + d_b(x,t).$$

Here, the second equality comes from Lemma 23 together with the fact that $t \in S_{b'}$ where $b'$ is the predecessor of $b$. Therefore,

$$d_b(s,t) = \min\{d_{b \circ a}(s,t), \min_{x \in S_b} d_{b \circ a}(s,x) + d_b(x,t)\}.$$

By the induction assumption, we have both

$$|\widehat{d}_{b \circ a}(s,t) - d_{b \circ a}(s,t)| \le \texttt{err}(b \circ a) \text{ and } |\widehat{d}_{b \circ a}(s,x) - d_{b \circ a}(s,x)| \le \texttt{err}(b \circ a).$$

Again by the fact that $t \in S_{b'}$ and $x \in S_b$, then $\texttt{IsShortcut}(x,t,b') = \textit{True}$ and therefore $|\widehat{d}_b(x,t) - d_b(x,t)| \le \zeta_2$. Combining these together we have

$$|d_b(s,t) - \widehat{d}_b(s,t)| \le \texttt{err}(b \circ a) + \zeta_2 \le \zeta_1 + (h - (|b| + 1))\zeta_2 + \zeta_2 = \texttt{err}(b).$$

**Case(2)** Suppose $\widehat{d}_b(s,t)$ is computed by the unique invocation of Algorithm 3 with parameter $k = 0$ and that $\texttt{IsShortcut}(s,t,b) = \text{False}$. Still, we assume $s, t \in V_{b \circ a}$ for $a \in \{0,1\}$. Since for any $z \in S_b$, both Recursive-APSD$(\mathcal{T}, G_{b \circ a}, (s,z), k')$ Recursive-APSD$(\mathcal{T}, G_{b \circ a}, (t,z), k')$ will be invoked with $k' > 0$, then from case (2), we have that for any $x, y \in S_b$, both $\widehat{d}_{b \circ a}(s,x)$ and

$\widehat{d}_{b \circ a}(y, t)$ is $\mathtt{err}(b \circ a)$ far from $d_{b \circ a}(s, x)$ and $d_{b \circ a}(y, t)$ respectively. Also, for any $x, y \in S_b$, since $\mathtt{IsShortcut}(x, y, b) = \textit{True}$, then

$$|\widehat{d}_b(x, y) - d_b(x, y)| \leq \zeta_2.$$

From the induction assumption, we also have

$$|\widehat{d}_{b \circ a}(s, t) - d_{b \circ a}(s, t)| \leq 2\mathtt{err}(b \circ a).$$

Combining these facts together Lemma 22, we have that

$$
\begin{aligned}
|\widehat{d}_b(s, t) - d_b(s, t)| &= |\min\{\widehat{d}_{b \circ a}(s, t), \min_{x, y \in S_b} \widehat{d}_{b \circ a}(s, x) + \widehat{d}_b(x, y) + \widehat{d}_{b \circ a}(y, t)\} \\
&\quad - \min\{d_{b \circ a}(s, t), \min_{x, y \in S_b} d_{b \circ a}(s, x) + d_b(x, y) + d_{b \circ a}(y, t)\}| \qquad (7) \\
&\leq 2\mathtt{err}(b \circ a) + \zeta_2 = 2\zeta_1 + 2(h - |b|)\zeta_2 - 2\zeta_2 + \zeta_2 \leq 2\mathtt{err}(b).
\end{aligned}
$$

This finishes the proof of Theorem 26. $\qquad \square$

## C   Proof of Theorem 9

To prove Theorem 9, we begin by presenting the decomposition procedure for $K_h$-minor-free graphs, which relies on the well-known lemma characterizing the separability of such graphs:

**Lemma 27** (Alon et al. [2]). *Let $h \in \mathbb{N}_+$ be an integer, and $G$ be a $K_h$-minor-free graph on $n$ vertices. Then, there exists a separator $S$ of $G$ of order at most $O(h^{3/2} n^{1/2})$ such that no connected components in $G(V \backslash S, E(V \backslash S))$ has more than $\frac{2}{3} n$ vertices.*

### C.1   Privacy Analysis

Based on the proof of Theorem 19, we give the following privacy guarantee on Algorithm 4.

**Theorem 28.** *Algorithm 4 preserves $(\varepsilon, \delta)$-differential privacy.*

The proof of Theorem 28 is almost identical to that of Theorem 19, except that we re-calibrate the variance of Gaussian noise according to the size of the $k$-covering set of a separator $S$, instead of the size of the whole separator.

### C.2   Utility Analysis

To demonstrate the improvement in finding $k$-covering sets within separators, we first present several useful facts and lemmas related to $k$-covering. The following lemmas state that each connected $n$-vertex graph has a $k$-covering set of size at most $O(n/k)$.

**Lemma 29** (Meir and Moon [40]). *Any connected undirected graph with $n$ vertices has a $k$-covering with size at most $1 + \lfloor n/(k+1) \rfloor$.*

**Lemma 30.** *Any graph with $n$ vertices composed of $x$ connected components has a $k$-covering with size at most $x + \lfloor n/(k+1) \rfloor$.*

*Proof.* Let $x$ be the number of components with sizes $n_1, n_2, \ldots, n_x$, respectively. Each component has a $k$-covering set of size $1 + \lfloor n_i/(k+1) \rfloor$) for $i = 1, 2, \ldots, x$. Summing these sizes yields the desired result. $\qquad \square$

We use the above lemmas to prove the following structural result, which establishes a covering set for the separator of $K_h$-minor-free graphs.

**Lemma 31** (Covering lemma for minor free graphs). *Fix an $h \geq 1$. Let $G = ([n], E)$ be a connected $K_h$-minor-free graph. Then for any $1 \leq d \leq n$, there exists a subset of vertices $S \subseteq [n]$ such that: (1) $S$ is a separator of $G$ and (2) there is a $d$-covering of $S$ with size at most $O(\sqrt{h^3 n/d})$.*

*Proof.* We first claim that for any connected graph on $n$ vertices, there exists a partition of $[n]$ into $s = O(n/d)$ disjoint subsets $(V_1, V_2, \cdots, V_s)$ such that the diameter of each $V_i (1 \leq i \leq s)$ is at

most $d$. Indeed, by Lemma 29, $G$ has a $d$-covering $\mathcal{C} \subseteq [n]$ of size $O(n/d)$. For each $u \in \mathcal{C}$, let $V'_u \subseteq [n]$ be the vertices covered by $u$ with at most $d$ hops. Then, the partition $(V_1, V_2, \cdots, V_s)$ can be constructed from $\{V'_u\}_{u \in \mathcal{C}}$ by removing duplicate elements. Next, by contracting each $V_i$ for $1 \leq i \leq s$ into a super-node and merging all multi-edges between the super-nodes, we obtain a smaller graph $\widetilde{G}$ with $s$ nodes, and clearly $\widetilde{G}$ is $K_h$-minor-free if $G$ is $K_h$-minor-free.

Thus, by the separation theorem for $K_h$-minor-free graphs (Lemma 27), $\widetilde{G}$ has a separator of size $O(h^{3/2}\sqrt{s}) = O(\sqrt{h^3 n/d})$. Picking one vertex from each super-node in the separator of $\widetilde{G}$ forms a $d$-covering set of the corresponding separator of the original graph, since each super-node has diameter at most $d$. This completes the proof of Lemma 31. □

**Remark 32.** We note that from Kuratowski's theorem [37], each planar graph is $K_5$-minor-free. Therefore, Lemma 31 directly implies that each connected $n$-vertex planar graph has a separator with a $d$-covering of size $O(\sqrt{n/d})$ for any $1 \leq d \leq n$.

**Lemma 33.** *For any two vertices $a$ and $b$ in a graph $X$, and $z_a, z_b \in Z$, where $Z$ is a $k$-covering of $X$, such that the hop distance between $a$ (resp. $b$) and $z_a$ ($z_b$) is at most $k$, we have:*

$$|d(a,b) - d(z_a, z_b)| \leq 2k \cdot W$$

*where $W$ is the maximum weight of edges in the graph.*

*Proof.* The lemma follows using the following set of inequalities: $|d(a,b) - d(z_a, z_b)| \leq |d(z_a, z_b) + d(a, z_a) + d(b, z_b) - d(z_a, z_b)| \leq 2k \cdot W$. □

The following lemmas are used to derive the error bound of the recursion based on two different types of separators.

**Lemma 34.** *Fix any $s, t \in V_b$, we have:*

$$d_b(s,t) = \min\{d_{b \circ 0}(s,t), d_{b \circ 1}(s,t)\}$$

*or*

$$d_b(s,t) \leq \min_{x,y \in S_b^k} (d_{b \circ 0}(s,x) + d_b(x,y) + d_{b \circ 1}(y,t)) + 2kW.$$

*Proof.* The proof is based on Lemma 33 and the proof of Lemma 22. Without loss of generality, assume that either both $s, t \in V_{b \circ 0}$, or $s \in V_{b \circ 0}$ and $t \in V_{b \circ 1}$ (otherwise, we switch $s$ and $t$, and the proof for $s, t \in V_{b \circ 1}$ is symmetric)

**1. When both $s, t \notin S_b^k$.** The proof is the same as the proof of the proof of Lemma 22.

**2. When at least one of $s, t$ is in $S_b^k$.** By the construction of $G_{b \circ a}^k$ such that

$$G_{b \circ a}^k = (V'_{b \circ a} \cup S_b^k, E(V_{b \circ a} \cup S_b^k) \backslash E(S_b^k)),$$

we have

$$\min_{x,y \in S_b^k} (d_{b \circ 0}(s,x) + d_b(x,y) + d_{b \circ 1}(y,t)) \leq \min_{x,y \in S_b} d_{b \circ 0}(s,x) + d_b(x,y) + d_{b \circ 1}(y,t) + 2kW$$

Combining Equation (3), Equation (4) and Equation (5) completes the proof in this case. □

**Lemma 35.** *Let $S_{b'}$ and $S_b$ be two adjacent separators, where $b = b' \circ u$ for $u \in \{0, 1\}$. Then for any $x \in S_b$, $a \in \{0,1\}^n$, and $y \in S_{b'}$, we have:*

$$d_b(x,y) \leq \min_{z \in S_b^k} d_b(x,z) + d_{b \circ a}(z,y) \leq d_b(x,y) + 2kW.$$

*Recall that $S_b^k$ is as defined in Step 4 of Algorithm 4.*

*Proof.* Let $P = (v_1 = x, v_2, \cdots, v_k = y)$ be any shortest path from $x$ to $y$, and let $w$ be the last vertex in $P$ such that $w \in S_b$. Such $w$ exists since $x \in S_b$. Then,

$$d_b(x,y) = d_b(x,w) + d_b(w,y) = d_b(x,w) + d_{b \circ a}(w,y) \geq \min_{z \in S_b} d_b(x,z) + d_{b \circ a}(z,y)$$

$$\geq \min_{z \in S_b^k} d_b(x,z) + d_{b \circ a}(z,y) - 2kW.$$

On the other hand,

$$d_b(x, y) \leq \min_{z \in S_b^k} d_b(x, z) + d_b(z, y) \leq \min_{z \in S_b^k} d_b(x, z) + d_{boa}(z, y)$$

due to the triangle inequality and the fact that $d_b(z, y) \leq d_{boa}(z, y)$. This completes the proof. □

Next, we write $\sigma_1' = c \cdot \sqrt{2 \log(1.25/\delta')}/\varepsilon'$ and $\sigma_2' = f(p, k) \cdot \sqrt{2 \log(1.25/\delta')}/\varepsilon'$.

**Lemma 36.** *Let* $h, p, k, c, \gamma$ *be as before and*

$$g(p, k, c, \gamma, h) := \sqrt{h + 3 \ln(\max\{f(p, k), c\}) + \ln\left(\frac{1}{2\gamma}\right)}.$$

*With probability at least* $1 - \gamma$, *the following holds for Algorithm 4:*

1. *For any* $s, t, b$ *where* $|b| = h$ *and that* `IsShortcut(`$s, t, b$`) =` True,
$$|d_b(s, t) - \widehat{d_b}(s, t)| \leq \sqrt{2}\sigma_1' \cdot g(p, k, c, \gamma, h).$$

2. *For any* $s, t, b$ *where* $|b| < h$ *and that* `IsShortcut(`$s, t, b$`) =` True,
$$|d_b(s, t) - \widehat{d_b}(s, t)| \leq \sqrt{2}\sigma_2' \cdot g(p, k, c, \gamma, h).$$

*Proof.* The proof is analogous to that of Lemma 25 in Appendix B. In particular, for any $s, t, b$ with `IsShortcut(`$s, t, b$`) =` True and $G_b$ is a leaf node, the difference in $\widehat{d_b}(s, t)$ and $d_b(s, t)$ is a Gaussian noise with variance $\sigma_1'$, and thus

$$\mathbf{Pr}\left[d_b(s, t) - \widehat{d_b}(s, t) \geq z\right] \leq \frac{\gamma}{m}$$

if we choose $z = \sqrt{2}\sigma_1'\sqrt{\log(m/(2\gamma))}$ for any $m \geq 1$ and $0 < \gamma < 1$. Also, if $G_b$ is not a leaf node, then according to Algorithm 4, we have

$$\mathbf{Pr}\left[d_b(s, t) - \widehat{d_b}(s, t) \geq \sigma_2'\sqrt{2 \log m/(2\gamma)}\right] \leq \frac{\gamma}{m}.$$

The number of noises added in Algorithm 4 is bounded by $m = 5 \cdot 2^h \cdot \max\{(f(p, k))^2, c^2\}$. Then, Lemma 36 can be proved by the union bound.

□

To prove the utility guarantee for Algorithm 4, we now state the following error bound which follows directly from Lemma 34, Lemma 35 and Lemma 36.

**Lemma 37.** *Fix any* $0 < \varepsilon, \delta < 1$, *and* $n, p \in \mathbb{N}$. *For any planar graph* $G = ([n], E, w)$ *that is* $(p, q, q')$-*recursively separable for some constants* $\frac{1}{2} \leq q \leq q' < 1$, *there is an* $(\varepsilon, \delta)$-*algorithm for estimating all-pair shortest distances in* $G$ *such that with high probability,*

$$|\widehat{d}(s, t) - d(s, t)| \leq O\left(\frac{f(p, k) \cdot \log^2 n \log(n/\delta)}{\varepsilon} + kW\right).$$

By applying Lemma 31 to bound $f(p, k)$ in Lemma 37, we are now ready to present the proof of the error bound stated in Theorem 9.

*Proof Of Theorem 9.* The privacy guarantee follows from Theorem 28. Now we give the proof for the utility guarantee.

(1) For a grid graph, let $p = \sqrt{n}$, the decomposition described in Section 3 gives that $f(p, k) = O(n^{1/4}\sqrt{W\varepsilon})$ when $k = n^{1/4}/\sqrt{W\varepsilon}$. Then, the result can be obtained by substituting this into Lemma 37. This choice of parameters is optimal due to the arithmetic-geometric mean inequality.

(2) For a $K_h$-minor-free graph, we choose $k = O(\frac{n^{1/3}}{(\varepsilon W)^{2/3}})$ to balance the number of hops required to reach the covering set for each vertex in the separator and the size of the $k$-covering of the separator. In this case, from Lemma 31, we have $f(p, k) = O(h \cdot (nW\varepsilon)^{1/3})$. Then, the desired result follows by substituting this into Lemma 37.

□

# D  The Complete Pseudocode of Algorithm 4

---

**Algorithm 4:** Constructing private shortcuts via sub-sampling in separator.

---

**Input:** Graph $G = (V, E, w)$, private parameter $\varepsilon, \delta$, sampling parameter $k$.

1. Recursively construct a binary tree $\mathcal{T}$ as described in this section.

2. Set $h = \log_{1/q'}(n/c)$, $\varepsilon' = \varepsilon/\sqrt{4h \log(1/\delta')}$, and $\delta' = \delta/(4h)$.

3. Let $\sigma = f(p, k)\sqrt{2\log(1.25/\delta')}/\varepsilon'$.

4. **for** *non-leaf node* $(G_b, S_b) \in V(\mathcal{T})$ **do**

    Find a $k$-covering set of $S_b$ and let it be $S_b^k$.

    **for** $x, y \in S_b^k$ *such that* $x \neq y$ **do**

        IsShortcut$(x, y, b) =$ *True*.

        Let $\widehat{d_b}(x, y) = d_b(x, y) + \mathcal{N}(0, \sigma^2)$.

    **end**

    **if** $b \neq \varnothing$ **then**

        Let $b'$ be the binary string that removes the last bit in $b$.

        Find a $k$-covering set of $S_{b'}$ and let it be $S_{b'}^k$.

        **for** $(x, y) \in S_{b'}^k \times S_b^k$ **do**

            **if** $x, y \notin S_{b'}^k \cap S_b^k$ **then**

                IsShortcut$(x, y, b') =$ *True*.

                Let $\widehat{d_b}(x, y) = d_b(x, y) + \mathcal{N}(0, \sigma^2)$.

            **end**

        **end**

    **end**

**end**

5. **for** *leaf node* $(G_b, \text{-}) \in V(\mathcal{T})$ **do**

    **for** $x, y \in V_b$ *such that* $x \neq y$ **do**

        IsShortcut$(x, y, b) =$ *True*.

        Let $\widehat{d_b}(x, y) = d_b(x, y) + \mathcal{N}\left(0, \frac{2c^2 \log(1.25/\delta')}{(\varepsilon')^2}\right)$.

    **end**

**end**

**Output:** The binary tree $\mathcal{T}$, and $\widehat{d_b}(u, v)$ for all $u, v \in V$ and $b \in \{0, 1\}^*$ such that
        IsShortcut$(x, y, b) =$ *True*.

---

# E  Empirical Simulation

Here, we present a numerical example over varying graph scales to validate our theoretical results and to conduct an empirical comparison between the utility of our algorithms and that of existing methods. In particular, for a given graph size $n \in \mathbb{N}_+$, we consider a $(\sqrt{\lceil n \rceil} \times \sqrt{\lceil n \rceil})$ grid graph, where edge weights are either uniformly sampled from a given range that can be defined as functions of the graph size. For comparison, we consider two previous algorithms that are specifically designed for grid graphs:

1. The algorithm by Sealfon [42] constructs an $O(n^{1/3})$-sized covering set for the entire grid graph, and then applies output perturbation to approximate the distances between every pair of points within the covering set.

2. Another approach is to first release all edge lengths using the Laplace mechanism. The length of a shortest path is then estimated using at most $O(n^{1/3})$ noisy edges, combined with the $k$-covering scheme from Sealfon [42] to eliminate the dependency on the edge weights. This idea was firstly introduced by Chen et al. [12].

To thoroughly examine the performance of these algorithms across different grid graphs, we designed two sets of empirical simulations shown in Figure 1: the low-weight scenario and the large-weight

scenario. In the low-weight scenario, each edge weight is uniformly sampled from $[0, 1]$, whereas in the large-weight scenario, each edge weight is sampled from $[0, n^{0.1}]$. In all experiments, we set $\varepsilon = 1$ and $\delta = n^{-10}$. Each test is repeated 50 times on MacBook M1 air, and the results are averaged.

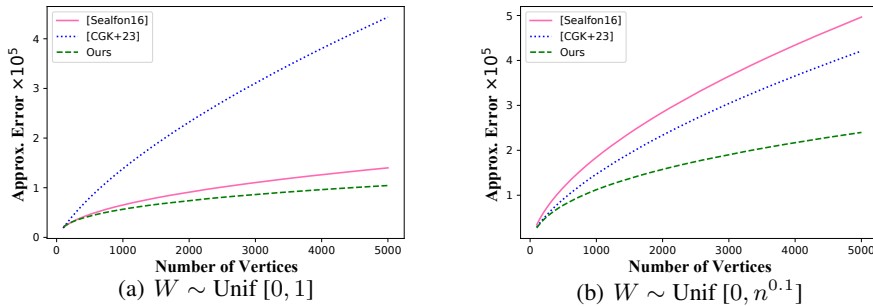

(a) $W \sim \text{Unif}\,[0, 1]$      (b) $W \sim \text{Unif}\,[0, n^{0.1}]$

Figure 1: Numerical simulations across various graph scales.

