# OpenReview forum: "A Generalized Binary Tree Mechanism for Private Approximation of All-Pair Shortest Distances"
_NeurIPS.cc/2025/Conference — NeurIPS 2025 poster_

### Official Review · Reviewer_HpTQ · 2025-06-23

**Clarity:** 3
**Significance:** 3
**Originality:** 3
**Rating:** 4
**Confidence:** 3

**Summary:**

This paper studies a fundamental differential privacy problem on graph, namely estimating all pairs shortest distances privately. The main contribution is to design a generic adaptation of the binary tree mechanism to a class of graphs that could be recursively separated, e.g., the grid graph. In particular, they show that if the graph is $(p, q, q')$ recursively separable, then their algorithm achieves an additive error of $p\log^3(n/\delta)/\epsilon$ for $(\epsilon, \delta)$-DP. For planar and general $K_h$ minor free graphs, their algorithm achieves an error of $\tilde O_\delta(h\cdot (\frac{(nW)^{1/3}}{\epsilon^{2/3}}))$ and the bound is further improved for grid graphs to $\tilde O_\delta(\frac{n^{1/4}W^{1/2}}{\epsilon^{1/2}})$.

**Questions:**

1. What do you think is the "correct" answer to DP APSP? At least from the results of this paper, it seems for a large class of graphs, the correct answer might be closer to the previous proved lower bound?

**Ethical Concerns:**

["NO or VERY MINOR ethics concerns only"]

**Final Justification:**

I'm happy with authors' response to my questions, in particular whether it's possible to generalize the technique to other problems. As my initial rating is positive, I've decided to keep it as is.

**Limitations:**

Authors didn't discuss limitations. One source of possible improvements is suggested in the weakness section.

**Paper Formatting Concerns:**

N/A.

**Quality:**

3

**Strengths And Weaknesses:**

Strengths:

1. The results are quite important, as it improves upon the additive error for a large class of graphs and it introduces the binary tree mechanism into the picture of DP APSP. It is known that binary tree mechanism works for path graph due to the range counting "1D" nature of APSP on a path. This paper demonstrates that for graphs that have good separability structures, you can essentially reduce to the case of a path (to some extent). While natural, this gives us better understanding for the class of graphs that could leverage the powerful binary tree mechanism.

2. Conceptually, the idea is quite neat and simple, and the paper is well-written. I find it easy to read and parse.

Weaknesses:

1. As illustrated in strengths, the main contribution of this paper is figuring out the correct class of algorithm that resembles the "path" behavior in order to apply binary tree mechanism. As such, once the property of recursively separable is spelled out, both the algorithms are the analysis become relatively straightforward. It does not bring much more insights beyond that. I think it would be interesting to abstract the general binary tree mechanism beyond DP APSP.

2. The paper is quite technically dense, authors could consider adding more discussions and a conclusion section in the appendix.

---

> ### Author Rebuttal · Authors · 2025-07-31
>
> We thank the reviewer for their valuable comments. We would love to add more discussions and high-level intuitions in the paper to make it less dense.
>
> > “I think it would be interesting to abstract the general binary tree mechanism beyond DP APSP.”
>
> This is indeed a very interesting point, and we would love to discuss more:
> The construction of our generalized binary tree mechanism relies on carefully designing the shortcuts inside and between separators, while adding shortcuts to reduce the number of hops in the shortest path between any two vertices is a natural strategy for APSD-type problems (with or without privacy constraints). Therefore, the construction of our framework is not problem-independent, and thus it is not very direct to abstract this beyond APSD. Moreover, our recursive strategies – such as leveraging triangle inequalities to control error rather than explicitly identifying the exact shortest path – are specifically tailored to address the challenges posed by general separable graphs in which the paths between two vertices are not unique, and the real shortest path may traverse separators in a nontrivial, back-and-forth manner. These properties also distinguish general recursively separable graphs from simpler structures like paths, in which the path between any pair of points is unique. Indeed, the Bentley&Saxe data structure as in the standard binary tree mechanism implicitly, but also crucially take advantage of the uniqueness of shortest paths, so that shortcuts can be easily constructed without further privacy loss.
>
>
> >”What do you think is the "correct" answer to DP APSP?”
>
> We share the same curiosity about this question, and the reviewer’s bringing it up has encouraged us to continue exploring how to close the gap in private APSD. Admittedly, we do not think we have a concrete answer. However, in light of our present results, we believe the “correct” dependency on $n$ might be different depending on whether we assume a bounded maximum edge weight.

---

> > ### Comment · Reviewer_HpTQ · 2025-08-01
> >
> > I want to thank authors for the detailed response, in particular on whether it's possible to abstract the general binary tree mechanism to other applications. I don't have further questions. As my initial rating is positive, I'll keep it as is.

---

### Official Review · Reviewer_Pxjc · 2025-06-29

**Clarity:** 4
**Significance:** 3
**Originality:** 3
**Rating:** 5
**Confidence:** 3

**Summary:**

The problem of releasing All-Pair Shortest Distances (APSD) on a (general) weighted graph with differential privacy has been well-studied. However, when the given graph has special properties, the error bound can be further improved. In this paper, the authors propose efficient algorithms with significantly improved bounds on graphs that are recursively separable (**Definition 1**). Typical examples of such graphs are trees or planar graphs. The property of recursive separability allows one to use the divide-and-conquer method (or a generalization of the binary tree mechanism) to recursively compute the shortest path, thus improving the bound for APSD on these graphs.

**Questions:**

No.

**Ethical Concerns:**

["NO or VERY MINOR ethics concerns only"]

**Final Justification:**

Thanks again for the rebuttal. I wish to see the open-source code of this paper in the final version.

**Limitations:**

Thanks for the submission. In its current form, I think the paper is well-written, and I cannot find limitations.

**Paper Formatting Concerns:**

No.

**Quality:**

4

**Strengths And Weaknesses:**

# Strengths

1. The proposed solutions seem reasonable and correct.
2. The paper is well-written, including intuitive explanations and sufficient theoretical analysis.

# Weaknesses

1. It would be better to open-source the implementations that can reproduce the results shown in Appendix E.

---

> ### Author Rebuttal · Authors · 2025-07-31
>
> We thank the reviewer for their valuable comments. We will include the link to the code in later non-anonymous versions of this paper.

---

> > ### Comment · Reviewer_Pxjc · 2025-08-01
> >
> > Thank authors for the response. I would like to see the source code in the revised version of the paper. I will keep my score as it is.

---

### Official Review · Reviewer_Ta62 · 2025-07-03

**Clarity:** 4
**Significance:** 3
**Originality:** 4
**Rating:** 5
**Confidence:** 4

**Summary:**

This paper presents an algorithm for solving the differentially private all-pairs shortest distances (APSD) problem on a special class of graphs called recursively separable graphs. The authors further develop an improved algorithm for a subset of these graphs, namely $K_h$​-minor-free graphs, and demonstrate enhanced approximation guarantees on specific graph classes. In particular, they achieve approximation error bounds of ${\tilde O}((nW)^{1/3}/\epsilon^{2/3})$ on planar graphs and ${\tilde O}(n^{1/4}W^{1/2}/\epsilon^{1/2})$on grid graphs.

A key technical innovation lies in a new recursive decomposition approach: the graph is partitioned into subgraphs using a divide-and-conquer strategy. For $K_h$​-minor-free graphs, the authors further construct a k-covering structure, which plays a crucial role in optimizing the algorithmic performance.

**Questions:**

The experimental evaluation is limited to randomly generated grid graphs. Could the authors test the algorithm on more diverse datasets or real-world graph structures to better demonstrate its practical performance?

**Ethical Concerns:**

["NO or VERY MINOR ethics concerns only"]

**Final Justification:**

The authors' detailed response answered my questions. Since my rating is positive, I will keep my rating.

**Limitations:**

This is a theoretical paper, and I do not see any negative societal impacts.

**Paper Formatting Concerns:**

I have no concerns regarding the formatting of the paper.

**Quality:**

4

**Strengths And Weaknesses:**

Strengths

1.The differentially private APSD problem is a classical and well-studied problem in the context of graph algorithms and privacy. Since its introduction by Sealfon[1], it has seen significant attention, and further progress in this area is of broad interest.

2.The proposed use of recursive structures over graphs introduces a novel and elegant algorithmic technique, showcasing theoretical creativity.

3.The algorithm achieves significantly improved approximation error bounds on widely studied graph families such as planar and grid graphs.

Weaknesses

1.The proposed method applies only to graphs with specific structural properties, such as recursively separable graphs and $K_h$-minor-free graphs (including planar and grid graphs). Its applicability to general graphs remains limited.

[1] Adam Sealfon. Shortest paths and distances with differential privacy. In Proceedings of the 35th ACM SIGMOD-SIGACT-SIGAI Symposium on Principles of Database Systems, pages 29–41, 2016.

---

> ### Author Rebuttal · Authors · 2025-07-31
>
> We thank the reviewer for their valuable comments.
>
> > “Could the authors test the algorithm on more diverse datasets or real-world graph structures to better demonstrate its practical performance?”
>
> We initially focus only on grid graphs mainly because the implementation of this case is much easier, and the goal is to verify whether the theoretical dependency on n is correct.
>
> To evaluate the performance of our algorithms on real-world graph structures, we consider metro line networks—one of the most natural application scenarios for private all-pairs shortest distance (APSD) computation. We note that there might be cycles in the metro network, so the binary tree mechanism is not directly applied.
> Specifically, we select metro networks from three Chinese cities: Jinan, Suzhou, and Nanjing. Each network is modeled as a planar graph based on the most up-to-date publicly available information, and we refer to these graphs as G1, G2 and G3 respectively. The datasets vary significantly in scale:
>
> Jinan Metro Network (G1) has 46 stations (including 2 interchange stations), 45 edges;
>
> Suzhou Metro Network (G2) has 134 stations (including 9 interchange stations), 138 edges;
>
> Nanjing Metro Network (G3) has 218 stations (including 20 interchange stations), 229 edges.
>
> (We must note that our dataset was manually compiled based on publicly available information and maps, so there may be minor inaccuracies.)
>
> We compare our proposed algorithm with two baselines: (1) the post-processing scheme from Sealfon [2016], and (2) the k-covering scheme in [Chen et al. 2023]. In all experiments, we set the privacy budget epsilon = 1 and delta = 0.01. Each graph is modeled based on the corresponding metro network topology, and as described in the main paper, edge weights are assigned independently at random from the uniform distribution over $[0, n^{0.1}]$, where n is the number of nodes in the graph.
> We define the error as the maximum difference in all-pair shortest distances, that is,
>
> $\text{error} = max_{u,v} | d_{u,v} - \hat{d}_{u,v} |$.
>
>
> Each result is the average over 10 independent trials. The results are summarized in the table below:
>
>
> | Algo    | Error in G1 (Jinan) | Error in G2 (Suzhou) | Error in G3 (Nanjing) |
> | -------- | ------- | ------- | ------- |
> | Sealfon 16  | 10.568 | 16.988 | 29.247 |
> | Chen et al. 23 | 12.860 | 15.793 | 22.772 |
> | Ours    | 14.118 | 16.416 | 19.972 |
>
> To better demonstrate the performance of our algorithm, we consider larger datasets. In particular, we consider the dataset of geographic faces from the dataset TIGER-Hawaii, provided by the U.S. Census Bureau. It corresponds to the state of Hawaii, with a total of 33,558 vertices, 80,882 edges and 17 connected components. This dataset considers five aggregation levels: states, counties, census tracts, census block groups and census blocks.
>
> To test the algorithm's performance at different scales while controlling the data size, we adopt the following strategy to select a random subgraph of size k that does not contain too many connected components: (1) uniformly and randomly select a vertex, (2) grow a subgraph using BFS strategy, truncate when the size of subgraph is already > k and (3) randomly select another vertex if the number of vertices found by the BFS is < k. Using the same settings as above, we present the following results under different sizes of the subgraph:
>
> | Algo    | k = 300 | k = 500 | k = 700 |
> | -------- | ------- | ------- | ------- |
> | Sealfon 16  | 26.040 |  32.245 | 40.441 |
> | Chen et al. 23 | 21.964 | 27.784 | 31.136 |
> | Ours    | 17.557 | 20.366 | 22.354 |

---

> > ### Comment · Reviewer_Ta62 · 2025-08-05
> >
> > Thank authors for the response. I will keep my score as it is.

---

### Official Review · Reviewer_PAPk · 2025-07-15

**Clarity:** 3
**Significance:** 3
**Originality:** 3
**Rating:** 5
**Confidence:** 3

**Summary:**

This paper proposes algorithms to compute all-pair distances in a
weighted graph under the constraints of differential privacy. Similar
to previous work, the algorithm proposed here are based on the binary
tree mechanism, a classical method to achieve good utility in the
continual observation model of differential privacy. In particular,
the algorithms proposed in this paper expand the class of graphs,
called recursively separable in the paper, to which this method is
applicable.

**Questions:**

Can you elaborate on which other interesting graph classes can be captured by recursively separable graphs?

Do you expect that one could design other optimized version of your framework for them?

**Ethical Concerns:**

["NO or VERY MINOR ethics concerns only"]

**Limitations:**

yes

**Quality:**

3

**Strengths And Weaknesses:**

Strengths

- the paper identifies a large class of graphs, recursively separable,
  that can be analyzed using the binary tree mechanism

- the paper presents algorithms for the compute all-pair distances
  problem that provide good utility on a large class of graphs

Weaknesses

- the improvement in utility for planar graphs and grid graphs do not
  follow directly from the main result but they require some further
  steps,

- the presentation of the algorithms can be improved with some more
  explanations and without showing the concrete algorithms.

The main contribution of the paper is to extend the range of
applicability of the binary tree mechanism for computing all-pair
distances in a weighted graph in a differentially private way.  In
particular, the paper interestingly connect it with a specific class
of graph which the paper calls recursively separable. Intuitively,
these are graphs that can be separated recursively in a balanced way.
This connection is intuitive and works well in practice providing a
general framework for the computing all-pair distances problem.  To
achieve good accuracy, the algorithm uses the topology of the graph to
identify private shortcuts that can be used in separators and across
differnet separators. These topology-based shortcuts permits to reduce
the error. Using this framework the paper can provide useful
algorithms for a large class of graphs, and achieve utility similar to
previous works on some classes of graphs (e.g. bounded tree-width
graphs).
However, in order to achieve improved utility with respect to previous
works on specific families of graphs, such as planar graphs and grid
graphs, the algorithmic framework needs some further optimization
(e.g. finding shortcuts in k-covering sets). This shows that there is
limiting trade-off between generality and accuracy of the proposed
framework. Nevertheless, the general framework and the optimizations
are both meaningful and provide good utility.
One minor negative aspect of the paper is in its presentation of the
algorithms. The paper tries to give some intuition about the algorithms
before presenting them formally. However, this intuition is limited
and a reader is left alone in decoding the algorithms. The presentation
could be improved by moving some of the formal details to the appendix,
leaving more space for the intuition behind them.
Also, as another minor negative point, some of the references are
missing important info. E.g. [1] has only title and year, [29] is just
an unpublished reference, [45] the journal title seems part of the
paper title.

---

> ### Author Rebuttal · Authors · 2025-07-31
>
> We thank the reviewer for their valuable comments.
>
> > “The presentation of the algorithms can be improved with some more explanations and without showing the concrete algorithms….One minor negative aspect of the paper is in its presentation of the algorithms…”
>
> We thank the reviewer for bringing this up; we will make the presentation of the algorithm more readable.
>
> > “Can you elaborate on which other interesting graph classes can be captured by recursively separable graphs?”
>
> Apart from planar graphs and grid graphs, other interesting recursively separable graphs can include (i) series–parallel graphs (with $p = 2$) that can be used in flow networks or circuit design and (ii) interval graphs (with p = their clique number minus one) that can be used in scheduling or resource reallocation. More generally, recursively separable graphs can be considered as a superclass of all bounded tree-width graphs.
>
> > “Do you expect that one could design other optimized version of your framework for them?”
>
> Yes. Since recursive separability is a relatively weak requirement, and we only look at minor-free graphs and grid graphs (with a more specified version of the algorithm for the latter), we would expect there are other optimized versions for other specific types of graphs. Presumably, one could take advantage of the extra structure, similar to our improved bounds for planar and minor-free graphs.

---

> > ### Comment · Reviewer_PAPk · 2025-08-04
> >
> > Thanks for your response which helps to clarify the scope of your contribution.

---

### Decision · Program_Chairs · 2025-09-17

**Decision:**

Accept (poster)

**Comment:**

The paper introduces a new method for privately computing the all pair shortest distances in a graph. Previously it was known how to do this for general graphs with error roughly sqrt(n)/epsilon, where n is the number of vertices and epsilon is the privacy parameter. The goal of the paper is to go beyond this bound for special families of graphs. It achieves this goal for "recursively separable" graphs, which include planar graphs and grid graphs as examples and only when the maximum weight is not large.

The strength of the result is that it is an approach to a broad family of graphs.

The downside is that the results require specific constructions for different subcases such as planar graphs and grid graphs and not a common algorithm, and it also imposes restrictions on the weight. It is also unclear if the bound given here are tight even for special cases.

All reviewers are positive about the contribution to a problem that attracted a lot of prior works in a broad class of graphs.